# Metabolomics Analysis Reveals the Influence Mechanism of Different Growth Years on the Growth, Metabolism and Accumulation of Medicinal Components of *Bupleurum scorzonerifolium* Willd. (Apiaceae)

**DOI:** 10.3390/biology14070864

**Published:** 2025-07-16

**Authors:** Jialin Sun, Jianhao Wu, Weinan Li, Xiubo Liu, Wei Ma

**Affiliations:** 1School of Pharmacy, Heilongjiang University of Traditional Chinese Medicine, Harbin 150040, China; klp15sjl@nefu.edu.cn (J.S.); 17515281505@163.com (J.W.); tyler2046@163.com (W.L.); 2Biological Science and Technology Department, Heilongjiang Vocational College for Nationalities, Harbin 150066, China

**Keywords:** *Bupleurum scorzonerifolium* Willd., mineral elements, metabolite profiles, growth regulation, growth years

## Abstract

It is difficult for the wild varieties of *Bupleurum scorzonerifolium* Willd. to meet the market demand. During its artificial cultivation, problems such as premature harvesting or delayed harvesting exist, resulting in a low yield and poor quality. This study aimed to explore the impact of different growth years on *Bupleurum scorzonerifolium* Willd. The researchers investigated the differences among 1- to 3-year-old *Bupleurum scorzonerifolium* Willd. and analyzed the plant’s metabolites. The results showed that with the accumulation of the growth cycle, the plant height and root–shoot ratio of three-year-old *Bupleurum* increased most significantly. With the accumulation of the growth cycle, the content of saponins in different parts of *Bupleurum* increased year by year and the content of saponins in three-year-old *Bupleurum* was the highest. Therefore, the three-year-old *Bupleurum* has the suitable harvesting period. This study identified key metabolites and related metabolic pathways. This achievement not only provides a theoretical basis for the establishment of a standardized cultivation technical system for *Bupleurum*, but also provides scientific support for guiding the efficient cultivation of *Bupleurum* in the field. At the same time, it has important theoretical value and practical significance for the protection, sustainable development and utilization of the resources of *Bupleurum* plants.

## 1. Introduction

*Bupleurum scorzonerifolium* Willd. is a perennial herbaceous plant of the genus *Bupleurum* in the Apiaceae family. Also known as red *Bupleurum*, it is mainly distributed in Northeast China, North China and other regions and is a commonly used medicinal plant [1]. Its dried roots are used as medicine. Clinically, it is commonly used to treat a variety of conditions such as cold and fever, irregular menstruation, distending pain in the chest and hypochondrium and organ prolapse [2,3,4]. Saikosaponin is the main active component of *Bupleurum*, mainly including saikosaponin A, C and D [5,6]. However, it is difficult for the wild resources of *Bupleurum scorzonerifolium* to meet the market demand and problems such as premature or delayed excavation exist in artificial cultivation, leading to a low yield and poor quality. As the content of bioactive components varies significantly under different growth years, determining the optimal growth years is crucial for ensuring the quality of this medicinal material.

Previous studies on *Bupleurum scorzonerifolium* have primarily focused on the chemical composition analysis of its roots and the pharmacological effects of saikosaponins, the main medicinal components. In terms of pharmacological effects, recent studies have confirmed that saikosaponins exhibit significant anti-inflammatory, antidepressant and antitumor activities [7,8,9,10]. Regarding chemical composition research, Zhu et al. used UHPLC-QTOF-MS to systematically analyze the chemical profiles of medicinal *Bupleurum* species from different regions. By comparing the chemical constituents between roots and aerial parts, they confirmed that the aerial parts cannot substitute for roots in medicinal use [11]. Yang et al. employed UPLC-Q-TOF-MS to conduct an in-depth comparative study on the chemical composition of four *Bupleurum* roots, successfully identifying 25 characteristic chromatographic peaks based on accurate mass data and fragmentation patterns [12]. Dong et al. integrated transcriptomic and metabolomic analyses to investigate the interspecific differences in the saikosaponin biosynthetic pathway among three *Bupleurum* species. Their research revealed that key genes such as CYP450, UGT and β-AS are closely associated with the biosynthesis of saikosaponins A, B1, C and D. Additionally, two P450 genes (Bc087391 and Bc036879) potentially involved in saikosaponin biosynthesis were identified, providing new insights into the molecular mechanisms underlying metabolic variations among different *Bupleurum* species [13].

The accumulation pattern of bioactive compounds in plants often shows a significant time trend with the growth years. Zhang et al. analyzed the correlation between flavonoid chemical components and the growth years of *Astragalus* membranaceus, determining that the growth years of Astragalus membranaceusare are the primary factors contributing to variations in its traits and chemical compositions [14]. Regarding *Codonopsis pilosula*, the content of active ingredients shows no significant difference between three-year-old and four-year-old plants; however, the active components in three-year-old *Codonopsis pilosula* remain relatively stable from September to November [15]. In *Polygonatum odoratum*, the content of active ingredients such as polysaccharides, flavonoids, and saponins peaks in the third year of growth and then stabilizes, with no statistically significant differences observed between the fourth and sixth years and the third year [16]. Evidently, the optimal growth years represent a crucial determinant of traditional Chinese medicine’s quality and notable variations exist across different plant species and growth durations. Metabolomics, a powerful tool for analyzing dynamic changes in metabolites within organisms, has been widely applied to the research of medicinal plants. By comprehensively deciphering the composition and content of metabolites, metabolomics can reveal the metabolic mechanisms underlying plant growth and development as well as the accumulation of active components. However, systematic studies on the dynamic changes of metabolites during the plant’s growth and development, as well as in-depth analyses of how the growth years affect metabolic pathways and the accumulation of medicinal components, are currently lacking. The accumulation patterns of bioactive compounds within plants often exhibit distinct temporal trends in relation to growth years. The marked variations in the concentrations of these active ingredients across different growth stages underscore the critical importance of precisely determining the optimal growth duration for medicinal plants. This parameter not only influences the efficacy of plant-derived pharmaceuticals, but also affects the overall quality and economic viability of medicinal plant cultivation [17,18].

In addition, mineral elements play a key role in plant growth and development as well as the accumulation of medicinal components [19,20,21,22]. For example, zinc, as a trace element with a relatively high demand in plants, is deeply involved in a series of important physiological activities and biochemical processes, such as cell differentiation, protein synthesis, photosynthesis and gene transcription [21]. Mg is not only a core component of chlorophyll, but also plays an indispensable role in photosynthesis and enzyme activation. Symptoms of a magnesium deficiency usually first appear in the old leaves and then gradually spread to the young leaves. Fe, as an essential element for maintaining the stability of the chloroplast structure and the normal operation of its functions, is decisive for chlorophyll synthesis and the improvement of the photosynthetic efficiency of plants [22]. An iron deficiency will seriously damage the ultrastructure and function of the chloroplasts [23]. However, the specific effects of the growth years on the metabolic profile and mineral element accumulation of *Bupleurum scorzonerifolium* remain unclear. Elucidating these mechanisms is of great significance for optimizing the cultivation techniques of this medicinal plant and determining the optimal harvest period.

The objectives of this study are as follows: (1) to evaluate the effects of different growth years on the yield and quality trait parameters of both the aboveground and underground parts of *Bupleurum*; (2) to reveal the impact of growth years on the content of saikosaponins in different tissues of *Bupleurum*; (3) to explore the influence of the growth years on the metabolite profiles of different tissues of *Bupleurum*; (4) to screen the key metabolites and metabolic pathways involved in the regulation of biosynthesis; and (5) to establish a visual model for the relationships among the growth years, metal content, saponin content and primary metabolism. The aim is to determine the optimal growth years for the artificially cultivated *Bupleurum* and provide a theoretical basis for the sustainable development and utilization of *Bupleurum* resources.

## 2. Materials and Methods

### 2.1. Experimental Instruments and Reagents

AR2130 electronic balance (Shanghai Shenglong Electronic Technology Co., Ltd., Shanghai, China); grinding instrument (MM400, Retsch GmbH, Haan, Germany); Milli-Q ultrapure water system (Millipore, Milford, MA, USA); RE532CS rotary evaporator (Shanghai Yalong Biochemical Instrument Factory, Shanghai, China); GL-16W benchtop centrifuge (Hunan Xiangyi Experimental Instrument Development Co., Ltd., Changsha, China); and Gas Chromatography–Mass Spectrometry (GC-MS): An Agilent 7890A GC (Agilent, Santa Clara, CA, USA) equipped with a non-polar DB-5 capillary column (30 m × 250 μm inner diameter, J&W Scientific, Folsom, CA, USA) is coupled with an Agilent 5975C quadrupole mass spectrometer (Agilent, USA). Analysis by Inductively Coupled Plasma Optical Emission Spectrometry (ICP-OES Optima 8000, PerkinElmer, Waltham, MA, USA); Hitachi High-performance Liquid Chromatograph L-2000 (Hitachi Ltd., Tokyo, Japan). Standard substances of sodium (Na), potassium (K), calcium (Ca), magnesium (Mg), copper (Cu), zinc [24], iron (Fe) and manganese (Mn) (purity ≥ 98%, Beijing Wanjia Shouhua Biotechnology Co., Ltd., Beijing, China). Standard substances of saikosaponin a, saikosaponin c and saikosaponin d (purity ≥ 98%, Sichuan Mansite Biotechnology Co., Ltd., Pengzhou, China); potassium fertilizer (Jinan Xinguan Chemical Products Co., Ltd., Jinan, China).

### 2.2. Plant Source

The experimental site is located in the *Bupleurum* Planting Research Base in Daqing, China (47°18′ N, 124°87′ E). It belongs in the continental monsoon climate in the North Temperate Zone, with a large temperature difference among the four seasons. The average annual temperature during the growing period from 2022 to 2024 is 4 °C, the average annual precipitation is 417.2 mm and the average annual sunshine duration is 2807 h. The altitude of the experimental site ranges from 142.7 to 152.4 m. The chemical properties of the soil are as follows: PH value is 7.78, the content of organic matter is 4.86 g/kg, the total nitrogen content is 1.64 g/kg, the available nitrogen content is 0.21 mg/kg, the available phosphorus content is 27.4 mg/kg and the available potassium content is 165.8 mg/kg. In the experimental site, one-year-old *Bupleurum* was planted in June 2023, two-year-old *Bupleurum* in June 2022 and three-year-old *Bupleurum* in June 2021. In August 2024, samples of *Bupleurum* are collected from the experimental fields with consistent geographical conditions, uniformity and on the same slope. August was chosen as it is within the optimal harvesting period for *Bupleurum* in the experimental area. At the time of collection, each *Bupleurum* specimen was directly segmented into four parts: roots, flowers, main shoots and lateral shoots. Every group gained three technical duplications and the weight and length of fresh products were measured at the same time. Simultaneously, a part of fresh *Bupleurum* was dried in a blast oven at 42 °C, and the remaining samples were stored at −80 °C.

### 2.3. Determination of the Saikosaponins Content

According to the provisions of the Chinese Pharmacopoeia [1], when *Bupleurum* is used as a medicinal herb, the total content of saikosaponin A and saikosaponin D in its medicinal part, the root, shall not be less than 0.3%. At the same time, another key medicinal active ingredient, saikosaponin C, is included in the scope of content determination. A total of 500 mg of every organ of the *Bupleurum* roots, flowers and main and lateral shoots was mixed into 25 mL of pure methanol (HPLC, ≥99.9%). The mixtures were ultrasonically treated and filtered at 30 °C for half an hour. The residue from the filter underwent two washes with 10 mL of methanol followed by merging and drying of the filtrates. The residue was re-dissolved in pure methanol (HPLC, ≥99.9%) in a 10 mL volumetric flask. A total of 20 μL was filtered for HPLC analysis (Hitachi, Tokyo, Japan). The chromatographic conditions were as follows (Table 1)—time: 0~50 min, acetonitrile: 25~90%, pure water: 75~10%; time: 50~55 min, acetonitrile: 90%, pure water: 10%. The chosen chromatographic column was a diamonsil C18 (4.6 mm × 250 mm, 5 μm) and the temperature was 25 °C, the flow rate was 0.8 mL∙min^−1^ and the calibration wavelength of the detector was 210 nm [6,25].

### 2.4. Elemental Analysis

First, 0.4 g of dried *Bupleurum* sample was transferred to a conical flask, followed by the addition of 5 mL concentrated nitric acid. Subsequently, the conical flask was placed on a hot plate. The heating process began at an initial temperature of 80 °C, which was then gradually increased to 150–180 °C for digestion. After digestion, the solution was subjected to an acid-removal process in a water bath. This continued until the volume of the solution decreased from 5 mL to 1 mL, completing the acid removal. The resulting solution was transferred to a volumetric flask and diluted to a fixed volume. Finally, it was filtered through a 0.45 μm membrane to obtain the sample mixture. The concentrations of Na, Ca, K, Mg, Mn, Zn, Fe and Cu were measured using Inductively Coupled Plasma Optical Emission Spectrometry (ICP–OES, Optima 8000, PerkinElmer, Waltham, MA, USA) [26]. During the analysis, the instrument automatically aspirated 3 mL of the prepared sample solution for determination.

### 2.5. GC-MS Analysis

The extraction and determination methods in this experiment refer to Sun [25]. Methanol and internal standard were added to plant samples for extraction. Then eddy current and ultrasound were applied to different parts of 60 mg extract. In addition, chloroform and water were added to the vortex and ultrasonic process, followed by centrifugation for 10 min to evaporate and dry the supernatant. The samples were incubated in a methoxyimino pyridine solution. After 90 min, the next step was oximation and derivatization. Then, 200 µL of methoxyamine hydrochloride pyridine solution (15 mg/mL) was added to the glass derivatization vial, vortexed for 2 min and placed it in a shaking incubator for oximation reaction for 90 min. Then, 200 µL of bis(trimethylsilyl)trifluoroacetamide BSTFA (containing 1% trimethylchlorosilane TMCS) derivatization reagent and 40 μL of n-hexane were added, vortexed for 2 min and reacted at 70 °C for 60 min. The sample was then taken out, left to stand at room temperature for 30 min, centrifuged at 10,000 rpm at low temperature for 10 min and was used for GC-MS analysis. Finally, the derivative solution was injected into the 7890A-5975C of Agilent. Temperature program: 0~125 °C, heating rate 8 °C/min; 125~210 °C, heating rate 4 °C/min; 210~270 °C, heating rate 5 °C/min; 270~305 °C, heating rate 10 °C/min; and final, 305 °C. The electron impact ion source was always kept at 260 °C and the voltage was constant at 70 V.

### 2.6. Multivariate Statistical Analysis

By hierarchical data clustering, the content values of each compound can be standardized to obtain a heat map of relative differences. We chose Graph Pad Prism 9.0 to draw the Graph. The associated network is visualized with Cytoscape version 3.7.1 so that the network is structured as a set of nodes (metabolites) and edges (connectors) between them. Based on Pearson correlation coefficient (r^2^ > 0.50 and FDR < 0.01), two nodes can be connected to each other.

## 3. Results

### 3.1. Comparative Analysis of Quality and Traits of Bupleurum Under Different Growth Years

*Bupleurum* is a perennial herbaceous plant. Its main root is often swollen and the stem is upright and green. It generally grows to a height of 30 to 80 cm. Taking Figure 1 as an example, the shoots and leaves of one-year-old *Bupleurum* are short and the root is relatively thin. The stem is single or grows in clusters of two to three. Two-year-old *Bupleurum* flowers once and its main root is relatively well-developed, conical in shape and reddish-brown in color. The root of three-year-old *Bupleurum* is thicker and the shoot is straight with many branches. The 1-year-old *Bupleurum* has not yet entered the flowering stage. The 2-year-old plants flower for the first time, with inflorescences in their umbel shape and light yellow petals. The 3-year-old plants show a significant increase in the flower quantity, along with an increase in the number of umbel rays and branching levels and an extension of the flowering period.

From Figure 2 and Table 2, it can be seen that the lengths of various parts of *Bupleurum* in different growth cycles increase with the accumulation of the growth cycle. Compared with one-year-old *Bupleurum*, the root, main shoot, lateral shoot and plant height of three-year-old *Bupleurum* increase by 3.64 times, 2.52 times, 7.56 times and 4.38 times, respectively. Among them, the lateral shoot has the largest increase and the main shoot has the smallest increase. When comparing two-year-old *Bupleurum* with one-year-old *Bupleurum*, the root, main shoot, lateral shoot and plant height increase by 3.02 times, 1.42 times, 6.87 times and 4.38 times, respectively. Similarly, the lateral shoot has the largest increase and the main shoot has the smallest increase. Compared with one-year-old *Bupleurum*, the root widths of three-year-old and two-year-old *Bupleurum* increase by 3.52 times and 0.91 times, respectively. The number of branches and the branch levels also increase with the accumulation of years. Overall, one-year-old *Bupleurum* has the lowest height for each part, the smallest root width and the fewest number of branches and branch levels. Two-year-old *Bupleurum* ranks second and three-year-old *Bupleurum* has the highest values for all parameters. Among them, the lateral shoot and the overall plant height of *Bupleurum* have the largest increases.

As shown in Figure 3 and Table 3, with the increase in the growth years, the dry matter mass of various organs of *Bupleurum* in different growth cycles shows a trend of increasing year by year. In two-year-old and three-year-old *Bupleurum*, the increase in the dry matter of the root is the most significant, while the increase in the lateral shoot is the smallest. Compared with one-year-old *Bupleurum*, the dry matter masses of the root, main shoot and lateral shoot of three-year-old *Bupleurum* have increased by 3.84 times, 2.14 times and 0.8 times, respectively; the dry matter masses of the root, main shoot and lateral shoot of two-year-old *Bupleurum* have increased by 1.95 times, 1.25 times and 0.63 times, respectively. In addition, this study also found that for both two-year-old and three-year-old *Bupleurum*, the accumulation of dry matter gradually decreases from the underground part to the above-ground part. This change is particularly obvious in the root and the main shoot and the increment of dry matter in three-year-old *Bupleurum* is significantly higher than that in two-year-old *Bupleurum*. In terms of reproductive growth, the 1-year-old *Bupleurum* plants have not yet entered the flowering stage. As the growth period extends, 2-year-old and 3-year-old *Bupleurum* show increasingly vigorous flowering. The dry matter mass of flowers from 3-year-old plants (7.87 ± 1.4 g) is significantly higher than that of 2-year-old flowers (3.456 ± 1.25 g). By the third year, the main shoots of *Bupleurum* are basically mature. The dry matter mass of the main branches in 3-year-old plants (3.754 ± 2.19 g) exhibits a less obvious increase compared to that of 2-year-old main branches (2.692 ± 0.97 g). In terms of the drying rate, there is no significant difference among *Bupleurum* of different growth years, which may be related to the water loss during the drying process. From the perspective of the root–shoot ratio (DSR), although the root dry weight of three-year-old *Bupleurum* is the largest, the significant growth of the above-ground parts such as the main shoot, lateral shoot and flowers leads to a lower root–shoot ratio compared with that of two-year-old and one-year-old *Bupleurum*.

### 3.2. Changes in Saikosaponins Content in Different Growth Years

By detecting the contents of saikosaponins in the roots, main shoots, lateral shoots and flowers of *Bupleurum* at different growth cycles (Figure 4 and Table 4), it is known that the overall accumulation patterns of saikosaponins A, C and D in different tissue parts present a relatively stable distribution. The content is the highest in the roots, followed by that in the flowers, and the lowest in the main shoots and lateral shoots. The distribution of saikosaponins within the *Bupleurum* plant exhibits significant tissue-specific differences. Among them, the root serves as the main organ for the enrichment of saikosaponins and its saikosaponin content is significantly higher than that of other above-ground parts such as the main shoot and lateral shoot, making it the core area for the accumulation of medicinal active ingredients. It is worth noting that although the flowers of *Bupleurum* belong to above-ground tissues, they show accumulation characteristics that are completely different from those of other above-ground parts and their saikosaponin content is significantly higher than that of the shoots, highlighting the special role of this part in the synthesis and storage of active ingredients. Analyzing from the perspective of the growth and development process, the saikosaponin content in various tissues of *Bupleurum* continues to increase with the growth years and the accumulation of saikosaponins in three-year-old plants reaches its peak.

### 3.3. Changes in Element Accumulation in Different Growth Years

The effects of different growth cycles on the contents of mineral elements in the main shoots, lateral shoots and roots of *Bupleurum* are shown in Table 5 and Figure 5, including four major elements (K, Ca, Na and Mg) and four trace elements (Mn, Cu, Zn and Fe). With the growth of *Bupleurum*, in the main shoot part, except for the fact that the contents of the Na and Ca elements increase slightly, the contents of the remaining elements decrease significantly. In the lateral shoot part, except for the element K, the contents of the elements in two-year-old *Bupleurum* are the lowest and the contents of the metals in one-year-old and three-year-old *Bupleurum* increase significantly. The elements in the roots of *Bupleurum* change the most compared with other parts. The contents of metals such as Na, Mn and Ca increase significantly. The contents of metals such as Cu and Fe decrease first and then increase, while the contents of K, Zn and Mg increase first and then decrease. In general, the accumulation of elements in the roots of *Bupleurum* is significantly higher than that in the lateral shoots and main shoots growing in the same year.

The principal component analysis (PCA) method was used to evaluate the degree of influence of the absorption and transportation of mineral elements in *Bupleurum* at different growth cycles. The total contribution rates of the principal components PC1 and PC2 of the main shoot were 67.97% and 31.90%, respectively (Figure 6A; Table 6). Most elements in the main shoot were concentrated in the negative interval, but Na and Ca were the main influencing elements in the main shoot and accumulated in the positive interval. Among them, the factor loading of Na was the largest in PC1, indicating that the first principal component mainly reflected the information of the Na element. And Ca had the largest factor loading in PC2, showing a highly positive correlation. According to the contribution rate of the factors to the total variance and combined with the factor loadings, Na (0.998) and Ca (0.889) were the characteristic elements of the main shoot part of *Bupleurum*.

The total contribution rates of the principal components PC1 and PC2 of the lateral shoot were 59.42% and 39.98%, respectively (Figure 6B; Table 6). All elements in the lateral shoot accumulated in the positive interval, among which Ca, Cu, Mg and Mn were the main influencing elements in the lateral shoot. The factor loading of Ca was the largest in PC1, indicating that the first principal component mainly reflected the information of the Ca element. Na had the largest factor loading in PC2 but showed a highly negative correlation. According to the contribution rate of the factors to the total variance and combined with the factor loadings, Ca (0.985) and Na (−0.945) were the characteristic elements of the lateral shoot part of *Bupleurum*.

The total contribution rates of the principal components PC1 and PC2 of the root were 61.88% and 37.97%, respectively (Figure 6C; Table 6). The trace elements in the root were mainly concentrated in the negative interval and the major elements were concentrated in the positive interval. The four elements of Na, Mg, Ca and Fe in the root were the main influencing elements in the root part during the growth cycle of *Bupleurum*. The contribution rates of different influencing factors in the root were lower than those in the main shoot and higher than those in the lateral shoot. The factor loading of Zn was the largest in PC1, indicating that the first principal component mainly reflected the information of the Zn element. Fe had the largest factor loading in PC2, showing a highly positive correlation. According to the contribution rate of the factors to the total variance and combined with the factor loadings, Zn (0.998) and Fe (0.873) were the characteristic elements of the root part of *Bupleurum*.

Figure 6D–F show the PCA results of *Bupleurum* in one-year, two-year and three-year growth cycles, respectively. It can be seen from the figure that with the extension of the growth cycle, the contribution rates of PC1 in the main shoot, lateral shoot and root were relatively stable at 67.97%, 59.42% and 61.88%, respectively, but the distribution intervals of the elements changed. The dominant role of characteristic elements (such as sodium in the main stem, calcium in the lateral stem and zinc in the root) is consistent in different years, but the load value of some elements (such as iron and manganese) may show a cumulative trend to attenuate the growth cycle, age-specific metabolism and the accumulation of saikosaponin.

### 3.4. GC-MS Analysis of Bupleurum in Different Growth Years

In order to understand the changes in the primary metabolites of *Bupleurum* at different growth cycles, we used GC-MS to identify the changes in the primary metabolites of *Bupleurum*. By comparing with our self-built database, a total of 59 primary metabolites of *Bupleurum* at different growth cycles belonging to 7 categories were identified (Figure 7A and Table 7), including 18 organic acids, 9 sugars, 4 alkyls, 8 polyols, 7 amino acids, 3 glycosides and 10 other substances. Cluster analysis was used to evaluate the relative differences in the accumulation patterns of *Bupleurum* at different growth cycles and different parts and then two main clusters were obtained (Figure 7B), with alkyls, glycosides and other substances in one cluster and organic acids, sugars, polyols and amino acids in the other cluster. Metabolites such as organic acids, sugars, polyols and amino acids accumulated in high abundance in the flowers and roots, while their abundances were lower in the main shoots and lateral shoots. With the accumulation of the growth cycle of *Bupleurum*, the abundances of substances such as organic acids, sugars, polyols and amino acids gradually decreased, while the abundances of alkyls, glycosides and other substances gradually increased. The abundance of the lateral shoots of two-year-old *Bupleurum* was the lowest and the abundance of the roots was the highest. The abundances of organic acids and glycosides in three-year-old *Bupleurum* were the lowest. Notably, the abundance of alkyls and glycosides increased by 32% and 25% from 1-year to 3-year samples, respectively, indicating their positive correlation with the growth years. In contrast, amino acid levels in 3-year-old roots decreased by 40% compared to 1-year-old roots and organic acid abundances in 3-year-old flowers dropped by 28%, reflecting a negative trend with the growth years. The changes in metabolites at different growth cycles reflect that the accumulation of metabolites during the growth and development process of *Bupleurum* is tissue-specific, with 3-year-old roots showing the highest accumulation of key medicinal components (alkyls and glycosides) and 2-year-old lateral shoots exhibiting the lowest overall metabolite abundance. This suggests that longer growth cycles may promote the biosynthesis of secondary metabolites, while shorter cycles favor primary metabolic activities supporting vegetative growth.

### 3.5. Metabolite Profiling of Bupleurum at Growth Cycles in Volcanic Map

Compared with two-year-old *Bupleurum*, a total of 22 metabolites in one-year-old *Bupleurum* showed significant changes (Figure 8A), among which 11 metabolites were significantly up-regulated (3 in the flowers, 2 in the lateral shoots, 1 in the main shoot and 5 in the roots) and 11 metabolites were down-regulated (2 in the flowers, 7 in the lateral shoots and 2 in the roots). The results of the KEGG enrichment analysis showed that in the comparison of 1 year vs. 2 years, the differential metabolites were mainly concentrated in the reductive carboxylic acid cycle (CO_2_ fixation), the citric acid cycle (TCA cycle), the biosynthesis of histidine and purine alkaloids and the biosynthesis of ornithine, lysine and nicotinic acid alkaloids (Figure 8B). When comparing two-year-old *Bupleurum* with three-year-old *Bupleurum*, approximately 29 metabolites changed (Figure 8C). A total of 22 metabolites (16 in the flowers, 1 in the lateral shoot, 1 in the main shoot and 4 in the roots) were up-regulated and 7 metabolites (4 in lateral shoots and 3 in main shoots) were down-regulated. For the comparison of 2 year vs. 3 years, the differences in the KEGG enrichment classification were also related to the following pathways: the insulin signaling pathway, the reductive carboxylic acid cycle (CO_2_ fixation), the citric acid cycle (TCA cycle), galactose metabolism, starch and sucrose metabolism and the biosynthesis of ornithine, lysine and nicotinic acid alkaloids (Figure 8D). When comparing one-year-old *Bupleurum* with three-year-old *Bupleurum*, 19 metabolites showed significant changes (Figure 8E), among which 9 metabolites (3 in the flowers, 3 in the lateral shoots, 1 in the main shoot and 2 in the roots) were up-regulated and 10 metabolites (3 in the flowers, 5 in the lateral shoots, 1 in the main shoot and 1 in the roots) were down-regulated. For the comparison of 1 year vs. 3 years, it involved the insulin signaling pathway, the reductive carboxylic acid cycle (CO_2_ fixation), the citric acid cycle (TCA cycle), the biosynthesis of histidine and purine alkaloids, galactose metabolism, starch and sucrose metabolism and the biosynthesis of ornithine, lysine and nicotinic acid alkaloids (Figure 8F). A Venn diagram was used to describe the common differential metabolites in different parts of *Bupleurum* at different growth cycles (Figure 8G). We found that nine differential metabolites (benzoic acid, cadaverine, galacturonic acid, citric acid, arabinofuranose, D-fructose, chlorogenic acid, ribitol and D-cellobiose), seven differential metabolites (L-rhamnose, benzylaminooctanol, arabinofuranose, D-fructose, chlorogenic acid, ribitol and D-cellobiose) and eight differential metabolites (succinic acid, malic acid, sucrose, arabinofuranose, D-fructose, chlorogenic acid, ribitol and D-cellobiose) existed in (2 year vs. 3 years) vs. (1 year vs. 2 years), (1 year vs. 2 years) vs. (1 year vs. 3 years) and (1 year vs. 3 years) vs. (2 year vs. 3 years), respectively. The abundances of six metabolites, namely arabinofuranose, D-fructose, chlorogenic acid, ribitol and D-cellobiose, showed obvious changes during the growth and development of *Bupleurum* at different growth cycles. The above six metabolites can be regarded as key metabolites for the growth and development of *Bupleurum*. The important metabolites of *Bupleurum* at different growth cycles were mapped to the reductive carboxylic acid cycle (CO_2_ fixation), the citric acid cycle (TCA cycle) and the biosynthesis of ornithine, lysine and nicotinic acid alkaloids to ensure metabolic regulation.

### 3.6. Establishment of PLS-DA Model of Primary Metabolites of Bupleurum in Different Growth Cycles

To reveal the differential metabolites among different parts of *Bupleurum* at various growth cycles, we further analyzed the 59 identified metabolites using the partial least squares-discriminant analysis (PLS-DA). The partial least squares method can explain the characteristics of four tissue parts (the flowers, lateral shoots, main shoots and roots), respectively. The difference in metabolites is the largest in the lateral shoots and the smallest in the roots. For the flower parts of *Bupleurum* at different growth cycles, PLS-DA-R2X[1] and PLS-DA-R2X[2] explain 24.9% and 14.9% of the characteristics of the metabolites, respectively (Figure 9A). In the lateral shoots of *Bupleurum*, PLS-DA-R2X[1] explains 29.2% of the characteristics of the metabolites and PLS-DA-R2X[2] explains 17.5% of the characteristics (Figure 9B). In the main shoots of *Bupleurum*, PLS-DA-R2X[1] explains 20.4% of the characteristics of the metabolites and PLS-DA-R2X[2] explains 13.2% of the characteristics (Figure 9C). For the root metabolites, PLS-DA-R2X[1] and PLS-DA-R2X[2] explain 18.6% and 21.4% of the characteristics of the metabolites, respectively (Figure 9D). In this study, the models for the flowers, lateral shoots and roots all show ideal effects, with Q^2^ > 0.5, and there is not a large gap between R^2^Y and Q^2^, indicating that the established models have good predictive abilities, are stable and reliable and can be used for the screening of differential metabolites (Table 8).

For *Bupleurum* of different ages, the growth years, differential metabolites, metal contents and saikosaponins of *Bupleurum* exhibit corresponding changes. In order to clarify whether these changes are correlated, based on the Pearson correlation coefficient, the growth cycles, metal contents, saikosaponin contents and metabolites were imported into Cytoscape to construct a visualized network model. All metabolites in the network are interrelated and the degree of correlation is indicated by generating lines between relevant nodes. These lines are encoded according to the evaluated positive (solid line) or negative (dashed line) correlations. The network includes four categories: the primary metabolism, metal contents, saikosaponin contents and different growth cycles (Figure 10). For one-year-old *Bupleurum*, there are 34 positively correlated and 33 negatively correlated substances in the grid diagram. Among them, it shows a significant positive correlation only with the contents of the metals Na, K, Ca and Mg in the roots, a significant negative correlation with saikosaponin SSc in the three parts and a significant positive correlation with saikosaponins SSa and SSd. It shows a significant negative correlation with primary metabolites such as ribofuranoside, sucrose, carboxylic acid, benzoic acid, succinic acid, galactonic acid, lactic acid, malic acid, palmitic acid, benzylaminooctanol, ribitol, cadaverine, barbital and glycerol and the rest are positively correlated. For two-year-old *Bupleurum*, there are 33 positively correlated and 37 negatively correlated substances in the grid diagram. Among them, it shows a significant negative correlation with all metals, a significant negative correlation with saikosaponins SSd-F, SSa-F and SSd-LS and the rest are positively correlated. It shows a significant negative correlation with primary metabolites such as galactofuranoside, cellobiose, L-rhamnose, ribofuranoside, 2-ethylbutyric acid, carboxylic acid, benzoic acid, succinic acid, chlorogenic acid, citric acid, galacturonic acid, lactic acid, malic acid, palmitic acid, 2,5-dimethyl-dihydroxybenzoate and cadaverine. For three-year-old *Bupleurum*, there are 53 positively correlated and 18 negatively correlated substances in the grid diagram. Among them, it shows a significant negative correlation with the metals Na-MS, Na-LS, Na-R, Mg-MS and Mg-LS and a positive correlation with all saikosaponins. It shows a significant negative correlation with primary metabolites such as arabinopyranose, galactofuranoside, D-glucose, glucopyranoside, sedoheptulose, ribofuranoside, 2-ethylbutyric acid and barbital.

## 4. Discussion

Growth years play a pivotal role in multiple facets of medicinal plants, including the accumulation of active ingredients, the development of the plant morphology, the regulation of physiological processes and the adaptation to environmental stress [17,27]. The results of this study indicate that with the increase in growth years, the lengths of various parts of *Bupleurum*, such as the roots, main shoots, lateral shoots and plant height, all show a significant increasing trend. Three-year-old *Bupleurum* has obvious advantages over one-year-old and two-year-old ones in these parameters. Its root length increased by 3.64 times, the main shoot grew by 2.52 times, the lateral shoots increased by as much as 7.56 times and the plant height increased by 4.38 times. In terms of the root width, three-year-old and two-year-old *Bupleurum* increased by 3.52 times and 0.91 times, respectively, compared with one-year-old *Bupleurum*. Moreover, the number of shoots and the branch levels also increased with the accumulation of years. This is consistent with the biological characteristics of *Bupleurum* as a perennial plant. The extension of the growth cycle provides a more sufficient time and material basis for its morphological construction. It is worth noting that the root–shoot ratio decreases with the increase in growth years. This phenomenon implies that during the growth process of *Bupleurum*, the growth of the above-ground part and the underground part does not occur synchronously, but there is a certain trade-off relationship. With the increase in growth years, the vigorous growth of the main shoots, lateral shoots and flowers in the above-ground part may consume more photosynthetic products and nutrients, thus affecting the material distribution of the underground roots and leading to a decrease in the root–shoot ratio.

The results of this study show that the contents of saikosaponin A, saikosaponin C and saikosaponin D in different parts of *Bupleurum* plants all exhibit an upward trend as the growth years of the plants increase. Moreover, the contents of these components in three-year-old *Bupleurum* plants reach the highest value. As a typical medicinal plant whose root is used for medicinal purposes, Panax ginseng has been the subject of the systematic determination of the ginsenoside content in various parts of Panax ginseng plants aged from 1 to 5 years in studies conducted by Liu et al. The research results show that the ginsenoside content exhibits a significant accumulation trend as the growth years increase, reaching its peak when the plants are 5 years old. Ginsenosides are mainly concentrated in the main root, followed by the lateral roots, and their contents in branches, petioles and leaves are relatively low [28]. Cui et al. employed gas chromatography–mass spectrometry (GC-MS) technology to conduct an in-depth analysis of the aroma characteristics of Panax ginseng at different growth stages. The study revealed that as the growth years of Panax ginseng increased, the contents of both ginsenosides and volatile oils in Panax ginseng showed a significant increasing trend, indicating a close correlation between the accumulation patterns of these two components and the growth cycle [29,30]. It is worth noting that the accumulation of active ingredients in plants is regulated by both the growth cycle and the geographical environment. Research shows that the content of active ingredients in plants not only changes dynamically with the number of growth years, but also exhibits significant differences due to environmental factors such as climate and soil in the planting areas. Therefore, the harvesting of medicinal plants does not simply follow the rule that “the longer the growth cycle, the better.” Instead, it is necessary to comprehensively consider multiple factors to determine the optimal harvesting period [31]. It has been found through research that *Bupleurum* not only abounds in saikosaponins in its roots, but also has this active ingredient detected in parts such as the main shoot, lateral shoots and flowers. Based on this, we speculate that the non-root tissues of *Bupleurum* plants also possess potential development and utilization values. In fact, in some regions of the southeastern part of our country, the whole *Bupleurum* plant has been used as medicine in local traditional medication habits. In Spain, people use the above-ground parts of *Bupleurum* as external preparations for medical purposes such as antibacterial and anti-inflammatory treatments. However, it is necessary to conduct further research to determine whether the saikosaponins in the roots and above-ground parts of *Bupleurum* are completely identical in terms of the substance types and functions.

Based on GC-MS-based untargeted metabolomics, we identified a total of 67 primary metabolites in *Bupleurum* at different growth cycles and in different tissue parts. Through PLS-DA analysis, six differential metabolites, namely Arabinofuranose, D-Fructose, Chlorogenic acid, Ribitol and D-Cellobiose were screened out as key metabolites for distinguishing the growth cycles of *Bupleurum*. As primary metabolites serve as synthetic precursors of secondary metabolites, the differences in metabolites may lead to changes in the metabolic pathways. Numerous previous studies have confirmed that differences in the growth cycle can have a significant impact on the accumulation of key metabolites and metabolic pathways in medicinal plants, which in turn determine their medicinal quality and efficacy [32,33,34]. Our findings revealed that *Bupleurum* plants of different growth years were engaged in multiple metabolic pathways, including the reductive carboxylic acid cycle, citric acid cycle and the biosynthesis of alkaloids from ornithine, lysine and niacin. The reductive carboxylic acid cycle and citric acid cycle, integral components of the carbon (C) metabolism, play a pivotal role in providing the energy and carbon skeletons essential for plant growth and development. Notably, the metabolisms of citric acid, malic acid and urea were suppressed, resulting in relatively low overall contents. This phenomenon might be attributed to decreased energy consumption, as well as altered conversion rates between C-metabolism and N-metabolism processes.

Plants need minerals for growth. Research by He and his colleagues has found that the concentration of mineral elements depends on the form of life and the plant parts. Different growth cycles have a significant impact on the element content within plants. Elements such as Ca, Mg, K, Na, Zn and Cu accumulate more in annual herbaceous plants than in perennial herbaceous plants among eleven herbaceous plant species [35]. This study found that, compared with the annual *Bupleurum*, the perennial *Bupleurum* accumulates more Na and Ca, while the contents of K, Mg, Zn, Cu, Fe and Mn are significantly lower. Secondly, different parts of the plant also have an influence on the elements. For instance, N, P, Mg and K are distributed in the stems and Fe and Al are distributed in the roots. It is likely because N, P, Mg and K are closely related to basic biological functions, such as photosynthesis, water utilization, respiration and reproductive allocation [36]. Previous studies have also confirmed that the demand of plants for specific nutrients is very important for promoting the healthy growth of plants and optimizing the yield [37,38,39]. In fact, the mineral elements in medicinal plants are closely related to the efficacy of their medicinal components [40]. Liu et al. explored the correlation between the mineral elements and total flavonoids in *Bupleurum* from different producing areas [41]. Xue et al. studied the relationship between the contents of saikosaponin A and saikosaponin D and the mineral elements within *Bupleurum chinense* plants. The research shows that copper positively regulates the accumulation of saikosaponin A and saikosaponin D, while sodium negatively regulates it [42]. There are significant differences in the content, accumulation amount and proportion of elements in *Bupleurum chinense*. For example, among the major elements, the content and accumulation amount of K are the highest, and among the trace elements, the content and accumulation amount of Fe are the highest [43]. Under the same germplasm and climatic conditions, soil becomes an important factor affecting the elemental content [44]. In addition to the properties of mineral elements, the germplasm of *Bupleurum* and soil conditions, it is also closely related to cultivation measures.

## 5. Conclusions

With the accumulation of the growth cycle, the height of three-year-old *Bupleurum* increased the most. Compared with one-year-old *Bupleurum*, the root length, main shoot height, lateral shoot height and plant height increased by 3.64 times, 2.52 times, 7.56 times and 4.38 times, respectively. The height increase of the two-year-old *Bupleurum* was the second. The root length, main shoot height, lateral shoot height and plant height increased by 3.02 times, 1.42 times, 6.87 times and 4.38 times, respectively. At the same time, the root width, the number of branches, the branching level and the dry matter weight of the plants also increased year by year. With the accumulation of the growth cycle, the content of saikosaponins in various parts of *Bupleurum* increased year by year and the content of saikosaponins in three-year-old *Bupleurum* was the highest. Using GC-MS non-targeted metabolomics, 59 primary metabolites were identified. Among them, six primary metabolites, namely arabinofuranose, D-fructose, chlorogenic acid, ribitol, D-cellobiose and phosphate, were key metabolites for the growth and development of *Bupleurum*. With the accumulation of the growth cycle of *Bupleurum*, the contents of organic acids, sugars, alcohols, amino acids, etc., gradually decreased, while the contents of alkyls, glycosides and other substances gradually increased. In the network diagram of one-year-old *Bupleurum chinense*, 34 positively correlated substances and 33 negatively correlated substances were involved. It showed a significant negative correlation with SSc and a significant positive correlation with SSa and SSd. In the network diagram of two-year-old *Bupleurum*, 33 positively correlated substances and 37 negatively correlated substances were involved. It showed a significant negative correlation with SSd-F, SSa-F and SSd-LS and the rest were positively correlated. In the network diagram of three-year-old *Bupleurum*, 53 positively correlated substances and 18 negatively correlated substances were involved and all of them showed a positive correlation with saikosaponins. Therefore, the three-year growth cycle of *Bupleurum* is considered to be the suitable growth cycle for its harvesting. This study systematically revealed for the first time the molecular mechanism by which the growth years affect the synthesis of saikosaponins in *Bupleurum* through regulating the primary metabolic pathway and innovatively constructed a four-dimensional regulation model of “growth years-element distribution-metabolic network-medicinal components”. This study found that by integrating transcriptomics and proteomics technologies, the dual regulatory mechanism of growth years in the processes of the primary metabolism and secondary metabolism can be deeply analyzed. This achievement not only provides a theoretical basis for the establishment of a standardized cultivation technical system for *Bupleurum chinense*, but also provides scientific support for guiding the efficient cultivation of *Bupleurum* in the field. At the same time, it has important theoretical value and practical significance for the protection, sustainable development and utilization of the resources of *Bupleurum* plants.

## Figures and Tables

**Figure 1 biology-14-00864-f001:**
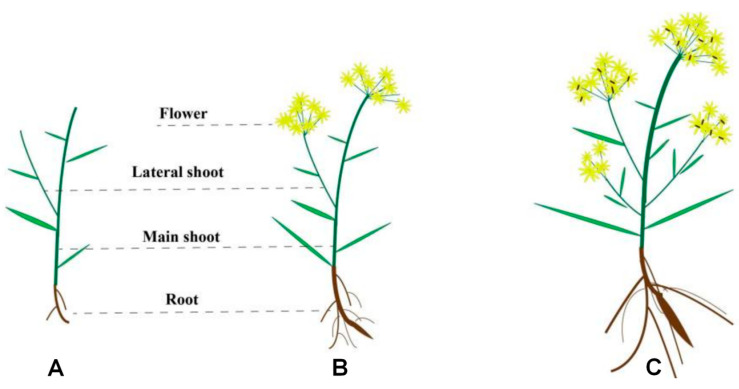
The appearance and morphology of *Bupleurum’s* different growth cycles. (**A**): One-year-old *Bupleurum*, (**B**): Two-year-old *Bupleurum*, (**C**): Three-year-old *Bupleurum*.

**Figure 2 biology-14-00864-f002:**
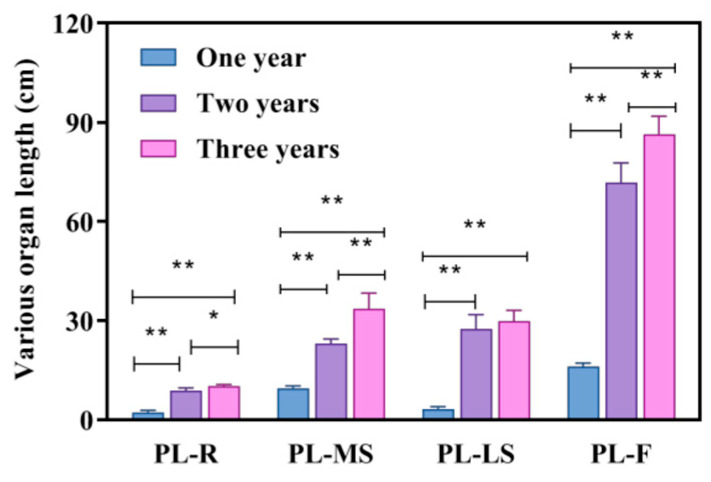
Growth of *Bupleurum* in different growth years. R: root, MS: main shoot, LS: lateral shoot, F: plant height; PL: length of each part of the plant. * and ** indicate *p* < 0.05 and *p* < 0.01, respectively.

**Figure 3 biology-14-00864-f003:**
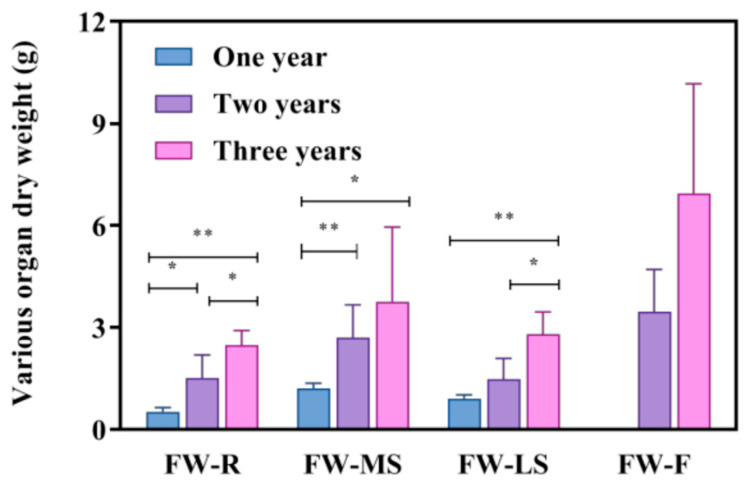
The amount of dry matter of *Bupleurum* in different growth years. R: root, MS: main shoot, LS: lateral shoot, F: flower, FW: fresh weight of each organ of *Bupleurum*. * and ** indicate *p* < 0.05 and *p* < 0.01, respectively.

**Figure 4 biology-14-00864-f004:**
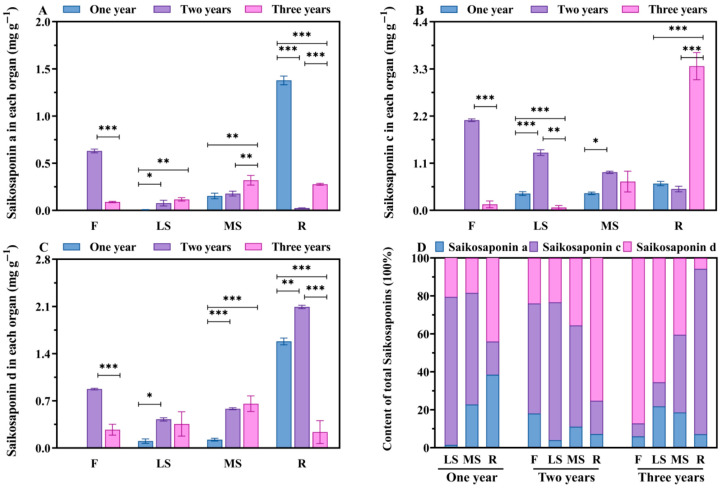
(**A**) The content of saikosaponin a in four parts. (**B**) The content of saikosaponin c in four parts. (**C**) The content of saikosaponin d in four parts. (**D**) The percentage of saikosaponin a, saikosaponin c and saikosaponin d content. R: roots; MS: main shoots; LS: lateral shoots; F: flowers. *, ** and *** signify significant *p* < 0.05, *p* < 0.01 and *p* < 0.001, respectively.

**Figure 5 biology-14-00864-f005:**
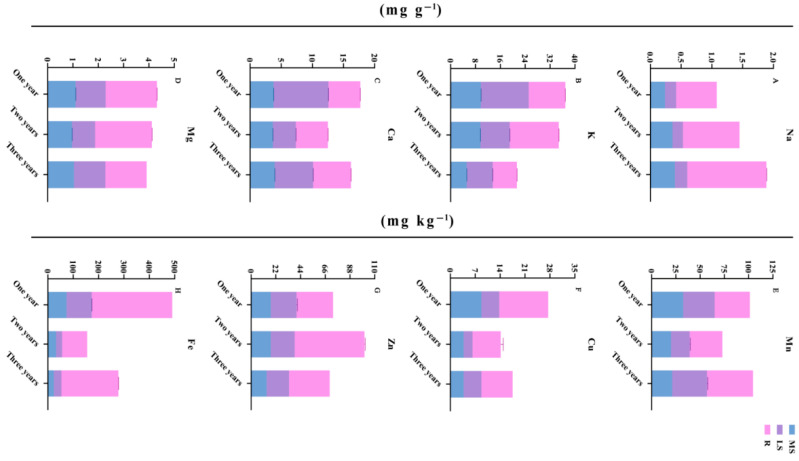
Changes in measured elements contents in *Bupleurum* during different growth cycles. One year: annual, two years: biennial, three years: triennial; R: root, MS: main shoot, LS: lateral shoot.

**Figure 6 biology-14-00864-f006:**
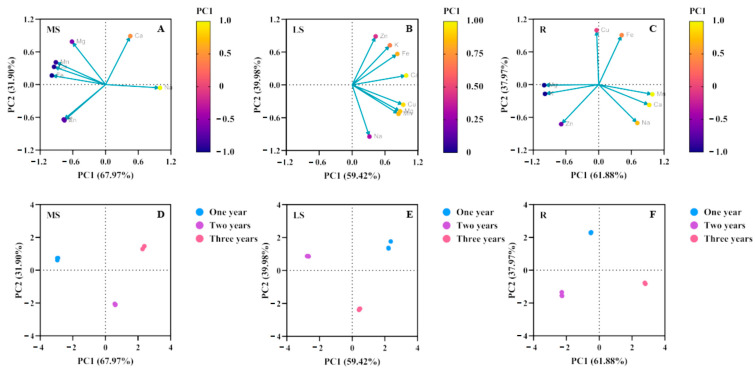
(**A**,**D**) Principal component contribution rate and load matrix of elements in main shoots across growth years. (**B**,**E**) Principal component contribution rate and load matrix of elements in lateral shoots across growth years. (**C**,**F**) Principal component contribution rate and load matrix of elements in roots across growth years. PC1 and PC2 indicate the first and the second principal component analysis, with yellow indicating a large contribution rate and purple indicating a small contribution rate.

**Figure 7 biology-14-00864-f007:**
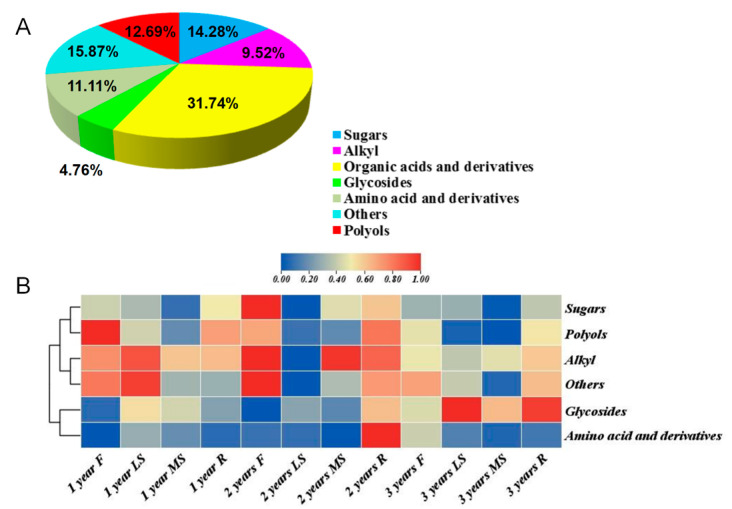
(**A**) Classification of 59 metabolites based on *Bupleurum*. (**B**) The cluster analysis and heat map for 59 metabolites with growth years and various parts. R: roots; MS: main shoots; LS: lateral shoots; F: flowers. The red and blue indicate up-regulated and down-regulated metabolites, respectively, while beige indicates an abundance of 0.

**Figure 8 biology-14-00864-f008:**
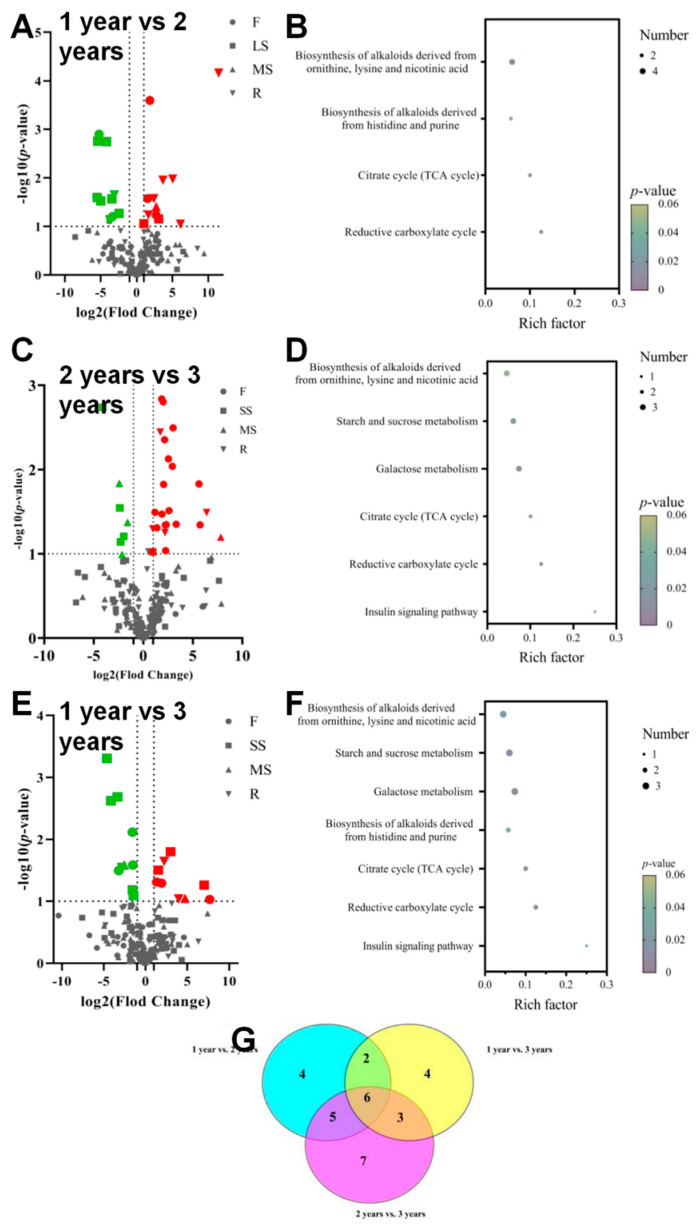
(**A**–**F**) Differential metabolite volcano plots and enrichment analysis results from Kyoto Encyclopedia of Genes and Genomes (KEGG). Red and green represent up-regulation and down-regulation of metabolic content, respectively. ●, ▇, ▲ and ▼ represent flower, main shoots, lateral shoots and roots, respectively. The *p*-value indicates the degree of enrichment and the closer the *p*-value is to 0, the more significant the enrichment is. The size of the dots represents the number of differential metabolites. (**G**) Venn diagram of key metabolites.

**Figure 9 biology-14-00864-f009:**
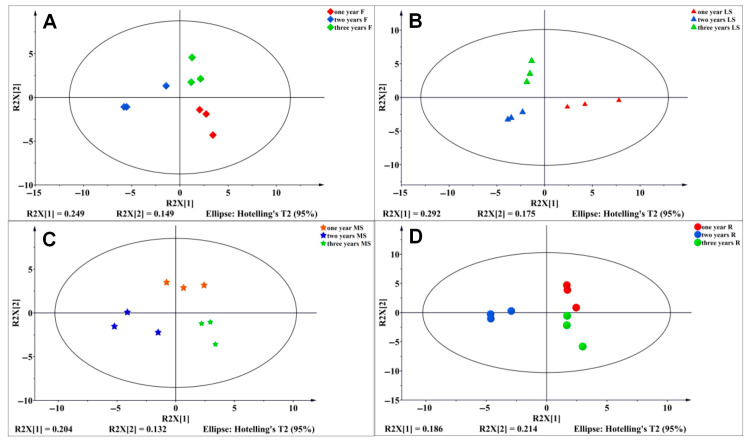
(**A**) Score diagram of *Bupleurum* in different growth cycles in flowers. (**B**) Score diagram of *Bupleurum* in different growth cycles in lateral shoots. (**C**) Score diagram of *Bupleurum* in different growth cycles in main shoots. (**D**) Score diagram of *Bupleurum* in different growth cycles in roots. ◆, ▲, ★ and ● represent F, LS, MS and R, respectively; red, blue and green represent annual, biennial and triennial *Bupleurum*, respectively.

**Figure 10 biology-14-00864-f010:**
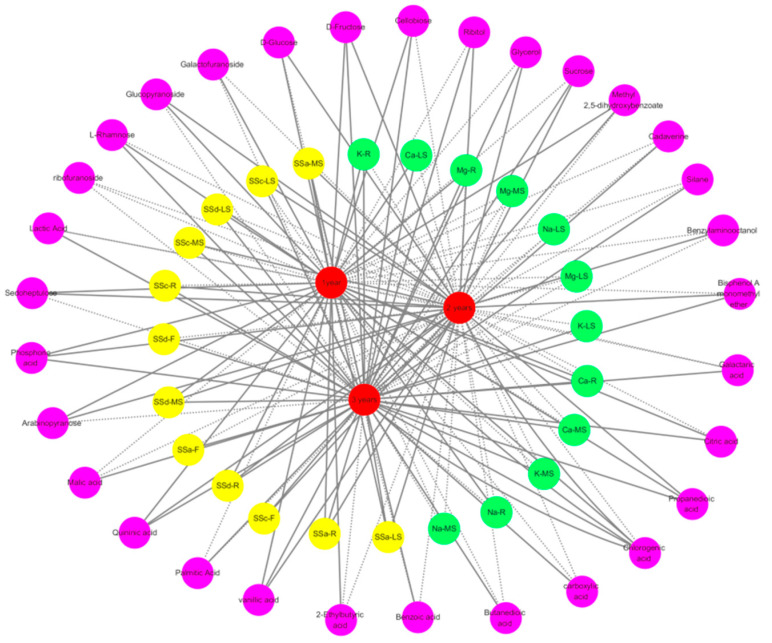
A correlation network diagram among different growth cycles of *Bupleurum*, saikosaponins, metals and differential metabolites. All substances are represented by circles. Among them, red nodes represent *Bupleurum* at different growth cycles, purple nodes represent primary metabolites, green nodes represent metal contents and yellow nodes represent the contents of saikosaponins in different parts of *Bupleurum*. The correlation is encoded by the Pearson coefficient. Positive correlations and negative correlations are indicated by solid lines and dashed lines, respectively, and the thickness of the lines represents the strength of the correlation.

**Table 1 biology-14-00864-t001:** Elution ratio.

Time	Acetonitrile (%)	Water (%)
0~10	25~38	75~62
11~20	38~51	62~49
21~30	51~64	49~36
31~40	64~77	36~23
41~55	77~90	23~10

**Table 2 biology-14-00864-t002:** Growth of *Bupleurum* in different growth years (*n* = 6).

	PL-R (cm)	PL-MS (cm)	PL-LS (cm)	PL-F (cm)	RW (cm)	NB	BL
One year	2.189 ± 0.63	9.538 ± 0.68	3.48 ± 0.51	16.068 ± 0.87	0.19 ± 0.01	3 ± 1	1–2
Two years	8.8 ± 0.84	23.1 ± 1.34	27.4 ± 4.39	71.8 ± 5.39	0.44 ± 0.05	7 ± 1	2–3
Three years	10.16 ± 0.50	33.6 ± 4.72	29.8 ± 3.27	86.4 ± 5.41	0.86 ± 0.18	13 ± 3	4–5

**Table 3 biology-14-00864-t003:** The amount of dry matter of *Bupleurum* in different growth years (*n* = 6), DR: The ratio of the weight of plants after drying to the weight before drying, DSR: ratio of dry weight of underground part to aboveground part of plant.

	FW-R (g)	FW-MS (g)	FW-LS (g)	FW-F (g)	DR	DSR
One year	0.51 ± 0.13	1.196 ± 0.17	0.9 ± 0.11	-	63.81% ± 9.71%	19.6% ± 4%
Two years	1.506 ± 0.68	2.692 ± 0.97	1.466 ± 0.62	3.456 ± 1.25	66.61% ± 13.25%	16.52% ± 3%
Three years	2.47 ± 0.44	3.754 ± 2.19	2.788 ± 0.66	7.87 ± 1.4	66.09% ± 12.2%	14.63% ± 10.3%

**Table 4 biology-14-00864-t004:** Saikosaponin content in different parts of *Bupleurum* in different growth periods.

	Saikosaponin A (mg·g^−1^)			Saikosaponin C (mg·g^−1^)			Saikosaponin D (mg·g^−1^)		
	One Year	Two Years	Three Years	One Year	Two Years	Three Years	One Year	Two Years	Three Years
F	-	0.629 ± 0.02	0.089 ± 0.01	-	2.104 ± 0.03	0.140 ± 0.08	-	0.874 ± 0.01	0.271 ± 0.08
LS	0.007 ± 0.01	0.078 ± 0.03	0.117 ± 0.02	0.391 ± 0.05	1.347 ± 0.07	0.067 ± 0.05	0.103 ± 0.03	0.425 ± 0.03	0.356 ± 0.18
MS	0.156 ± 0.03	0.178 ± 0.02	0.319 ± 0.05	0.398 ± 0.03	0.888 ± 0.026	0.673 ± 0.242	0.124 ± 0.02	0.582 ± 0.02	0.657 ± 0.116
R	1.378 ± 0.01	0.024 ± 0.01	0.278 ± 0.01	0.625 ± 0.05	0.491 ± 0.06	3.364 ± 0.32	1.581 ± 0.05	2.092 ± 0.02	0.235 ± 0.17

**Table 5 biology-14-00864-t005:** Contents of measured metal elements of *Bupleurum* with different growth years (*n* = 3).

Element	Treatment	MS	LS	R	RDR
Na (mg/g^−1^)	One year	0.24 ± 0.0007	0.18 ± 0.0007	0.66 ± 0.0031	61.13%
	Two years	0.36 ± 0.0007	0.17 ± 0.0000	0.92 ± 0.0019	63.51%
	Three years	0.40 ± 0.0013	0.20 ± 0.0007	1.28 ± 0.0029	68.27%
K (mg/g^−1^)	One year	9.72 ± 0.0610	15.39 ± 0.0087	11.80 ± 0.0102	31.96%
	Two years	9.62 ± 0.0323	9.39 ± 0.0477	15.80 ± 0.0824	45.40%
	Three years	5.18 ± 0.0425	8.37 ± 0.0170	7.78 ± 0.0454	36.47%
Ca (mg/g^−1^)	One year	3.77 ± 0.0335	8.81 ± 0.0118	5.12 ± 0.0025	28.89%
	Two years	3.66 ± 0.0188	3.70 ± 0.0207	5.15 ± 0.0293	41.17%
	Three years	3.96 ± 0.0325	6.17 ± 0.0254	6.07 ± 0.0333	37.48%
Mg (mg/g^−1^)	One year	1.10 ± 0.0038	1.19 ± 0.0014	2.02 ± 0.0056	46.84%
	Two years	0.96 ± 0.0022	0.91 ± 0.0012	2.24 ± 0.0094	54.54%
	Three years	1.04 ± 0.0014	1.24 ± 0.0036	1.63 ± 0.0007	41.68%
Mn (mg/g^−1^)	One year	32.50 ± 0.0000	32.50 ± 0.0000	36.25 ± 0.0000	35.80%
	Two years	20.00 ± 0.0000	19.17 ± 0.7217	33.75 ± 0.0000	46.29%
	Three years	21.25 ± 0.0000	35.83 ± 0.7217	47.50 ± 0.0000	45.42%
Cu (mg/g^−1^)	One year	8.75 ± 0.0000	5.00 ± 0.0000	13.75 ± 0.0000	50.00%
	Two years	3.75 ± 0.0000	2.50 ± 0.0000	7.92 ± 0.7217	55.88%
	Three years	3.75 ± 0.0000	5.00 ± 0.0000	8.75 ± 0.0000	50.00%
Zn (mg/g^−1^)	One year	17.50 ± 0.0000	22.92 ± 0.7217	32.50 ± 0.0000	44.57%
	Two years	17.50 ± 0.0000	21.25 ± 0.0000	62.08 ± 0.7217	61.57%
	Three years	13.75 ± 0.0000	20.00 ± 0.0000	36.25 ± 0.0000	51.79%
Fe (mg/g^−1^)	One year	73.75 ± 0.0000	99.17 ± 0.7217	317.50 ± 0.0000	64.74%
	Two years	32.50 ± 0.0000	23.75 ± 0.0000	98.75 ± 0.0000	63.71%
	Three years	23.75 ± 0.0000	30.00 ± 0.0000	224.17 ± 0.7217	80.66%

RDR: Root distribution ratio.

**Table 6 biology-14-00864-t006:** Principal component contribution rate and load matrix of elements.

MS	PC1	PC2	LS	PC1	PC2	R	PC1	PC2
Na	0.998	−0.060	Na	0.316	−0.945	Na	0.498	−0.843
K	−0.767	−0.640	K	0.688	0.722	K	0.884	−0.391
Ca	0.453	0.889	Ca	0.985	0.171	Ca	0.696	0.664
Mg	−0.617	0.787	Mg	0.874	−0.485	Mg	0.973	−0.132
Mn	−0.912	0.409	Mn	0.846	−0.531	Mn	0.573	0.798
Cu	−0.946	0.325	Cu	0.931	−0.364	Cu	0.952	−0.251
Zn	−0.754	−0.656	Zn	0.429	0.886	Zn	0.998	−0.019
Fe	−0.986	0.166	Fe	0.824	0.565	Fe	0.266	0.873
Total variation explained	67.97%	31.90%		59.42%	39.98%		61.88%	37.97%

**Table 7 biology-14-00864-t007:** Classification of 59 primary metabolites of *Bupleurum*’s different growth cycles.

Classification	Metabolites	Classification	Metabolites
Acids and derivatives (18)	2-Propenoic acid	Sugars (9)	D-Mannose
	Aminobutanoic acid		D-Cellobiose
	Aminocyclopentanecarboxylic acid		Arabinofuranose
	Benzoic acid		D-Fructose
	Butanedioic acid		D-Glucose
	Butyric acid		L-Rhamnose
	Carboxylic acid		α-D-Lactose
	Chlorogenic acid		Sedoheptulose
	Citric acid		Sucrose
	Lactic acid	Polyols (8)	D-Mannitol
	Galactaric acid		Glycerol
	Palmitic acid		Muco-Inositol
	Pentanedioic acid		Cuminyl alcohol
			Benzenediol
	Pipecolic acid		Benzylaminooctanol
	Propanedioic acid		Ribitol
	Quininic acid		Phenol
	β-D-Glucofuranuronic acid	Alkyl (4)	Heptane
	Malic acid		
Glycosides (3)	α-D-Galactopyranoside		
	α-D-Glucopyranoside		Nonane
	α-D-Ribofuranoside		Sulfone
Others (10)	Ether		Decane
	Galactose oxime		
	Glycerol monostearate	Amino acid and derivatives (7)	L-Alanine
	Methyl benzoate		L-Valine
	Heptabarbital		Urea
	Isoquinolinium		Amide
	Androst		Ethanolamine
	Carbamate		Cadaverine
	Cyclohexene		Diaziridine
	Monopalmitin		

**Table 8 biology-14-00864-t008:** PLS-DA model parameters of *Bupleurum* GC-MS data with different growth cycles. Q^2^ (cum): Evaluate the prediction ability of the model through cross-validation and measure the prediction accuracy of the model for new data. The closer the value is to 1, the stronger the prediction ability of the model. Generally, Q^2^ > 0.5 indicates that the model has good prediction ability. R^2^Y-Q^2^: This value can reflect whether there is over-fitting phenomenon in the model. If the value is small, the fitting effect and prediction ability of the model are more balanced.

	R^2^X (cum)	R^2^Y (cum)	Q^2^ (cum)	R^2^Y-Q^2^
PLS-DA(F)	0.397	0.814	0.134	0.680
PLS-DA(LS)	0.467	0.893	0.518	0.375
PLS-DA(MS)	0.337	0.850	0.019	0.831
PLS-DA(R)	0.751	0.999	0.900	0.099

## Data Availability

The original contributions presented in this study are included in this article; further inquiries can be directed to the corresponding authors.

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
