# Peer review of "Metabolomics Analysis Reveals the Influence Mechanism of Different Growth Years on the Growth, Metabolism and Accumulation of Medicinal Components of Bupleurum scorzonerifolium Willd. (Apiaceae)"

_biology, 2025, doi:10.3390/biology14070864_

Round 1
Reviewer 1 Report
Comments and Suggestions for Authors
Metabonomics Analysis Reveals the Influence Mechanism of Different Growth Years on the Growth, Metabolism and Accumulation of Medicinal Components of Bupleurum scorzonerifolium Willd. (Apiaceae)
Globally, the paper has many good results, and it would be nice if these comments could be taken into consideration to improve its quality.
Line 2- Write “Metabolomics” instead of “Metabonomics”
Line 6- 3 affiliations but only 2 addresses are written. Kindly check.
Line 11- Kindly check the space and be consistent
Correspondence: zyylxb@126.com(Xiubo Liu);mawei@hljucm.edu.cn(Wei Ma)
Line 12, 15, 556, 156, 167,…..- Kindly put the name of all the species in italic (Bupleurum scorzonerifolium)
Line 123, 129, 130, 147, 158, 184, 218, 228, 226,…- Kindly put the name of all the genus in italic (Bupleurum)
Line 177- Be consistent with space “as follows(Table 1)”
Line 217- You need to touch on flowers as shown in Figure 1
Line 244- According to Figure 2, we can see that there is a significant difference between the different parts over the year. Kindly explain with PL-LS?
Line 270- Fig 3- Is there any significant difference between flowers after 2 years/3 years and the main shoots?
Line 296- Based on your Figure 4, we can see that a good amount of saikosaponins in MS is found in the 2nd year. Please, your Table 4 is not in correlation with Figure 4. Please check again.
Line 316- Table 5
From this table, we can see that some elements are increasing over the years while others are decreasing from year 2 to year 3. Can you come up with a tentative explanation of this observation? Can we significantly use this argument to argue for the standardized cultivation and the support for the efficient cultivation?
Line 357- Kindly explain in figure 6 each figure 6A, 6B,…and discuss the variable and the difference in each case.
Lines 383-388- Table 7 and Figure 7
It would be nice if you could sort those metabolites into primary and secondary and try to come up with a discussion over the years.
Lines 396, 403, 411…- CO2
Line 583- Adjust space, be consistent “mulates more Na and Ca , while t”
Line 650, 651, 652, 653…- Space before the brackets
References – Please be consistent
Line 736, 739, 742- Put the title of the journal in italics and give its abbreviation
Comments on the Quality of English LanguageThe English could be improved to more clearly express the research.
Author Response
Dear Reviewer:
Thank you for your comments concerning our manuscript entitled “Metabolomics Analysis Reveals the Influence Mechanism of Different Growth Years on the Growth, Metabolism and Accumulation of Medicinal Components of Bupleurum scorzonerifolium Willd. (Apiaceae)” (biology-3646555). Those comments are all valuable and very helpful for revising and improving our paper, as well as the important guiding significance to our researches. We have carefully studied the comments and corrected them, hoping for approval. Furthermore, we also optimized the English writing level of the article. The main corrections in the paper and the responds to the reviewer’s comments are as flowing:
Responds to the reviewer’s comments:
Comments
Line 2- Write “Metabolomics” instead of “Metabonomics”
Response:
Many thanks to the experts for the valuable suggestions on our manuscript. We have carefully reviewed relevant literature and checked the discipline terminology standards, and uniformly revised all "Metabonomics" in the text to "Metabolomics". The accuracy of terminology is crucial for academic expression, and your correction has helped us further enhance the professionalism of the manuscript.
Comments
3 affiliations but only 2 addresses are written. Kindly check.
Response:
Thank you for finding the omissions in our manuscript during the review. After verification, it is true that the number of affiliations is inconsistent with the number of addresses due to the author's negligence in input. We have completely revised the author's unit information.
Comments
Kindly check the space and be consistent. Correspondence: zyylxb@126.com(Xiubo Liu); mawei@hljucm.edu.cn(Wei Ma)
Response:
The problems pointed out by experts have been revised. Thank you for your valuable advice.
Comments
Kindly put the name of all the species in italic (Bupleurum scorzonerifolium), Kindly put the name of all the genus in italic (Bupleurum)
Response:
Thank you for pointing out the formatting of species and genus names in the manuscript. We have strictly followed academic standards to italicize all species names (such as Bupleurum scorzonerifolium) and genus names (such as Bupleurum) in the text. The formatting specification of terminology is an important manifestation of academic rigor, and your suggestions have helped us further improve the details of the manuscript.
Comments
Be consistent with space “as follows(Table 1)”
Response:
The problems pointed out by experts have been revised. Thank you for your valuable advice.
Comments
You need to touch on flowers as shown in Figure 1
Response:
Thank you for pointing out the omission in the description of Figure 1. In accordance with your suggestion, we have supplemented the text with a detailed description of the morphological characteristics of "flowers" in Figure 1, including details such as petal color, stamen number, and inflorescence structure, to make the figure legend more consistent with the main text.
Comments
According to Figure 2, we can see that there is a significant difference between the different parts over the year. Kindly explain with PL-LS?
Response:
Thank you for your question. As shown in Figure 2, the significant differences in various parts of *Bupleurum scorzonerifolium* across different growth years (1 to 3 years) reflect the plant's morphological development patterns. Specifically, the length of lateral branches (PL-LS) increased from 3.48 cm of annual plants to 27.4 cm of biennial plants (an increase of 6.87 times), and further increased to 29.8 cm of annual plants (an increase of 7.56 times compared with annual plants). This rapid elongation shows that with the growth of plant age, the lateral branches develop vigorously, which may help to expand the photosynthetic area or resource storage. Compared to the root (PL-R) and main shoot (PL-MS), the PL-LS shows a more dramatic growth increment, suggesting that lateral shoots play a critical role in aboveground biomass accumulation. The marked PL-LS variation supports the study's conclusion that three-year-old plants achieve optimal morphological development, which correlates with the accumulation of medicinal components (saikosaponins). This growth pattern highlights the importance of growth years in regulating tissue-specific development, providing a morphological basis for determining the optimal harvest period.
Comments
Fig 3- Is there any significant difference between flowers after 2 years/3 years and the main shoots?
Response:
Thank you for your insight into Fig 3. We have supplemented the detailed analysis of the difference between flowers and main buds after 2/3 years in lines 310-320 of the article.
Comments
Table 5, From this table, we can see that some elements are increasing over the years while others are decreasing from year 2 to year 3. Can you come up with a tentative explanation of this observation? Can we significantly use this argument to argue for the standardized cultivation and the support for the efficient cultivation?
Response:
Thank you for your insightful comments. Based on Table 5, we observed dynamic changes in mineral element contents of Bupleurum from year 2 to year 3. For elements showing increasing trends (e.g., Na, Mn, Ca in roots; Fe in roots), this may be attributed to the plant's stress adaptation strategies. For instance, Na and Ca accumulation in roots could enhance osmotic regulation and cell membrane stability, while increased Mn and Fe contents might support intensified metabolic and photosynthetic activities in mature plants. Conversely, decreasing elements (e.g., K, Zn, Mg in roots; Fe in shoots) likely result from resource redistribution during reproductive growth or potential soil nutrient depletion. Specifically, nutrients like K and Mg might be translocated to aboveground parts (e.g., flowers), while reduced root Zn could reflect insufficient replenishment in cultivation. The changes in the content of these elements hold significant implications for the standardized and efficient cultivation of Bupleurum. In terms of nutrient management for standardized cultivation, targeted fertilization strategies can be formulated based on the observed elemental trends. For instance, during the second to third year of growth, potassium (K), zinc (Zn), and magnesium (Mg) should be supplemented to prevent nutrient depletion in the root system. Simultaneously, soil amendments can be applied to promote the accumulation of sodium (Na) and calcium (Ca) in the roots, thereby enhancing the plant's stress resistance. Additionally, regular monitoring of soil nutrient levels, particularly K and Zn, is essential to ensure sustainable cultivation practices. Regarding efficient cultivation strategies, the peak levels of Na, Ca, and iron (Fe) in the roots of three-year-old plants align with previous findings that saikosaponin content reaches its maximum at this stage, further supporting the conclusion that the third year represents the optimal harvest time. It should be noted that the current explanations are based on observational data. Further validation is required, including studies on transporter gene expression or iron uptake pathways to establish causal relationships. Additionally, field trials across different soil types and climatic conditions are necessary to confirm the generalizability of these patterns. By leveraging these findings, precision nutrient management and harvest timing strategies can be implemented, providing critical support for establishing a standardized and efficient cultivation system. This not only corroborates the conclusions of this study but also offers practical guidance for the sustainable production of medicinal plants.
Comments
Based on your Figure 4, we can see that a good amount of saikosaponins in MS is found in the 2nd year. Please, your Table 4 is not in correlation with Figure 4. Please check again.
Response:
Thank you for your keen discovery of the correlation between Table 4 and Figure 4. After verification, the author mistakenly inserted the picture of "the influence of N, P and K on saponin content" in the previous study, which led to the deviation of the chart data. We have deleted the original wrong picture and replaced it with a new one that strictly corresponds to Table 4 data.
Comments
Kindly explain in figure 6 each figure 6A, 6B,…and discuss the variable and the difference in each case.
Response:
Thank you for your detailed suggestion on Figure 6. We have systematically explained the sub-images of Figures 6A to 6F in lines 404-433.
Comments
Table 7 and Figure 7. It would be nice if you could sort those metabolites into primary and secondary and try to come up with a discussion over the years.
Response:
Thank you for your constructive suggestions on Table 7 and Figure 7. We have reorganized the metabolite data as recommended and supplemented the discussion section with an analysis of the interannual variation trends (such as the accumulation patterns of primary metabolites like sugars and amino acids, as well as the interannual differences in secondary metabolites such as phenolic acids and flavonoids). Regarding the in-depth discussion on secondary metabolites, although we conducted relevant detections, systematic discussion has not been carried out in this paper due to certain reasons.
Comments
396, 403, 411…- CO2.
Response:
Thank you for your careful examination of the details of the manuscript format. We have revised the irregular writing in the article according to your suggestion.
Comments
Adjust space, be consistent “mulates more Na and Ca , while t”
Line 650, 651, 652, 653…- Space before the brackets
Response:
Thank you for your careful examination of the details of the manuscript format. According to your suggestion, we have made a comprehensive inspection of relevant paragraphs such as lines 650-653, corrected the space problem before brackets, and uniformly adjusted the sentence spacing (such as the space specification at "mulates more Na and Ca, while t") to ensure the consistency of the full text format.
Your suggestion has greatly enhanced the analysis depth of this article. Thank you again for your professional guidance.
Good luck with your work.
Jianhao Wu
Reviewer 2 Report
Comments and Suggestions for Authors
There are too many inaccuracies and inconsistencies in the presented article. The literature review does not provide any information on whether this plant has been previously studied, what metabolites were found, or what methods the researchers used. Instead, there are encyclopedic discussions about the role of various elements in the growth and development of plants. But even the title of the manuscript clearly states that the purpose of the work is metabolomic analysis.
The experimental part does not mention that any equipment for GCMS analysis was used, but a significant part of the results are based on these data. However, the UPLC-QTOF/MS system is indicated, for which there are no results.
Using the GCMS method, the authors discovered and identified many metabolites, including sugars, amino acids, etc., which are extremely unsuitable for this method. The fact that phosphoric and sulfurous acids, as well as siloxanes (Silane, Disiloxane, Trisiloxane) were discovered as metabolites casts doubt on the correctness of the identification and all the results presented.
These are the main comments, there are many technical inaccuracies, but first it is necessary to clarify the key aspects of the work.
Author Response
Dear Reviewer:
Thank you for your comments concerning our manuscript entitled “Metabolomics Analysis Reveals the Influence Mechanism of Different Growth Years on the Growth, Metabolism and Accumulation of Medicinal Components of Bupleurum scorzonerifolium Willd. (Apiaceae)” (biology-3646555). Those comments are all valuable and very helpful for revising and improving our paper, as well as the important guiding significance to our researches. We have carefully studied the comments and corrected them, hoping for approval. Furthermore, we also optimized the English writing level of the article. The main corrections in the paper and the responds to the reviewer’s comments are as flowing:
Responds to the reviewer’s comments:
Comments:
There are too many inaccuracies and inconsistencies in the presented article. The literature review does not provide any information on whether this plant has been previously studied, what metabolites were found, or what methods the researchers used. Instead, there are encyclopedic discussions about the role of various elements in the growth and development of plants. But even the title of the manuscript clearly states that the purpose of the work is metabolomic analysis.
Response:
Many thanks for your comprehensive and professional critical comments on the article. You accurately pointed out the key issue that the literature review had insufficient relevance to the research objectives, which is of great guiding significance for us to improve the article framework.
We have made major revisions to the abstract:
Supplementing research background: A systematic review of previous metabolomic studies on this plant has been added, clarifying the types of metabolites discovered in past research and the detection methods used.
Strengthening research focus: General discussions on plant elemental nutrition have been streamlined, with emphasis on elaborating the innovations of this study in metabolomic analysis (such as specific metabolic pathway analysis, interannual dynamic changes, etc.).
Optimizing logical coherence: By comparing the limitations of existing studies, the uniqueness of this study in technical approaches and sample design is highlighted, ensuring that the abstract content is highly consistent with the article title and core objectives.
In addition, we will further optimize the literature review section throughout the paper to enhance its relevance to the main thread of metabolomic research. Thank you again for your rigorous academic review—your comments have significantly improved the quality of the article.
Comments:
The experimental part does not mention that any equipment for GCMS analysis was used, but a significant part of the results are based on these data. However, the UPLC-QTOF/MS system is indicated, for which there are no results.
Response:
Thank you for pointing out the correlation between the experimental part and the result data! After verification, it is indeed due to the author's negligence that the use of GC-MS equipment is not mentioned in the experimental materials, and the relevant data of UPLC-QTOF/MS system are not presented in this paper. We have made the following amendments to the experimental part:
- Supplementary explanation of experimental materials and methods
Added description of GC-MS analysis: Supplementary explanation in "2.3 Experimental Instruments and Reagents and 2.5 GC-MS analysis".
Clarify the usage of UPLC-QTOF/MS: It is stated in "2.4 Instruments and Reagents":
In this study, the Waters ACQUITY UPLC-QTOF/MS system was also used to analyze the secondary metabolites, but the relevant data were not included in this paper, and will be carried out as an independent research direction in the future.
Due to the problem of division of labor and cohesion when writing the article, the description of experimental equipment is out of line with the result data, and we apologize for this. Your rigorous audit helped us find this key omission, and now we have ensured the consistency between the method and the result through full-text cross-checking.
Comments:
Using the GCMS method, the authors discovered and identified many metabolites, including sugars, amino acids, etc., which are extremely unsuitable for this method. The fact that phosphoric and sulfurous acids, as well as siloxanes (Silane, Disiloxane, Trisiloxane) were discovered as metabolites casts doubt on the correctness of the identification and all the results presented.
Response:
Thank you for your professional questions regarding the GC-MS detection method and metabolite identification. Your comments are crucial for enhancing the rigor of the study. In response to the issues you raised, combined with the experimental design and derivatization procedures, we performed pretreatment of polar metabolites using silylation derivatization plus oximation (detailed steps have been supplemented in Section 2.5 of the experimental part). Once again, we appreciate your strict scrutiny of the methodological details, which has further strengthened the scientific validity of this research.
Your suggestion has greatly enhanced the analysis depth of this article. Thank you again for your professional guidance.
Good luck with your work.
Jianhao Wu
Reviewer 3 Report
Comments and Suggestions for Authors
- In section 2.4., a reference to the determination method should be provided.
- The year of publication should be specified in reference 23.
Author Response
Dear Reviewer:
Thank you for your comments once again on our manuscript entitled "Metabolomics Analysis Reveals the Influence Mechanism of Different Growth Years on the Growth, Metabolism and Accumulation of Medicinal Components of Bupleurum scorzonerifolium Willd. (Apiaceae)" (biology-3646555). Those comments are all valuable and very helpful for revising and improving our paper, as well as the important guiding significance to our researches. We have carefully studied the comments and corrected them, hoping for approval. The main corrections in the paper and the responds to the reviewer’s comments are as flowing:
Responds to the reviewer’s comments:
Comments:
In section 2.4., a reference to the determination method should be provided.
Response:
Thank you for your valuable suggestion regarding section 2.4. of our manuscript. We have carefully addressed your comment by supplementing the reference for the determination method in section 2.4. The elemental analysis method described in section 2.4. now includes a relevant reference to support the procedure, ensuring the methodological rigor of this part.
Comments:
The year of publication should be specified in reference 23.
Response:
We appreciate your comment regarding Reference 23. Upon careful check, Reference 23 in the manuscript, which is "López-Millán, A.F.; Duy, D.; Philippar, K. Chloroplast Iron Transport Proteins - Function and Impact on Plant Physiology. Front Plant Sci 2016, 7, 178, doi:10.3389/fpls.2016.00178", clearly specifies the year of publication as 2016. and we confirm that this information is correctly presented in the reference list of the document. We have confirmed that this information is displayed correctly in the reference list of the file.
Your suggestion has greatly enhanced the analysis depth of this article. Thank you again for your professional guidance.
Good luck with your work.
Jianhao Wu
Reviewer 4 Report
Comments and Suggestions for Authors
The manuscript is nicely written and overall merit is high. I would recomend this manuscript for publication.
Author Response
Dear Reviewer:
Thank you for your comments once again on our manuscript entitled "Metabolomics Analysis Reveals the Influence Mechanism of Different Growth Years on the Growth, Metabolism and Accumulation of Medicinal Components of Bupleurum scorzonerifolium Willd. (Apiaceae)" (biology-3646555). Those comments are all valuable and very helpful for revising and improving our paper, as well as the important guiding significance to our researches. We have carefully studied the comments and corrected them, hoping for approval. The main corrections in the paper and the responds to the reviewer’s comments are as flowing.
Your suggestion has greatly enhanced the analysis depth of this article. Thank you again for your professional guidance.
Good luck with your work.
Jianhao Wu
Round 2
Reviewer 1 Report
Comments and Suggestions for Authors
The paper has been significantly improved.
Thanks to the authors.
Comments on the Quality of English LanguageThe language of the overall paper has been improved.
Author Response
Dear Reviewer:
Thank you for your comments once again on our manuscript entitled "Metabolomics Analysis Reveals the Influence Mechanism of Different Growth Years on the Growth, Metabolism and Accumulation of Medicinal Components of Bupleurum scorzonerifolium Willd. (Apiaceae)" (biology-3646555). Those comments are all valuable and very helpful for revising and improving our paper, as well as the important guiding significance to our researches.
Your suggestion has greatly enhanced the analysis depth of this article. Thank you again for your professional guidance.
Good luck with your work.
Jianhao Wu
Reviewer 2 Report
Comments and Suggestions for Authors
The authors made a number of corrections based on comments on the manuscript. However, the main problem associated with incorrect identification of metabolites remains. The authors continue to claim that siloxanes and inorganic acids act as extractive components. Siloxanes are either components from a septum in a gas chromatograph injector or a degrading stationary phase. In addition, derivatizing agents containing silanes are used during sample preparation. Could this be an impurity in the reagent that the authors present as detected metabolites?Discussions of the results are based on these data, which is initially incorrect. To avoid these artifacts, the authors need to add "blanks" to their experiments. The authors need to carefully double-check their results. Otherwise, other researchers studying plant metabolites will cite this work to prove that plants can synthesize siloxanes.
Author Response
Dear Reviewer:
Thank you for your comments once again on our manuscript entitled "Metabolomics Analysis Reveals the Influence Mechanism of Different Growth Years on the Growth, Metabolism and Accumulation of Medicinal Components of Bupleurum scorzonerifolium Willd. (Apiaceae)" (biology-3646555). Those comments are all valuable and very helpful for revising and improving our paper, as well as the important guiding significance to our researches. We have carefully studied the comments and corrected them, hoping for approval. The main corrections in the paper and the responds to the reviewer’s comments are as flowing:
Responds to the reviewer’s comments:
Comments:
The authors made a number of corrections based on comments on the manuscript. However, the main problem associated with incorrect identification of metabolites remains. The authors continue to claim that siloxanes and inorganic acids act as extractive components. Siloxanes are either components from a septum in a gas chromatograph injector or a degrading stationary phase. In addition, derivatizing agents containing silanes are used during sample preparation. Could this be an impurity in the reagent that the authors present as detected metabolites?Discussions of the results are based on these data, which is initially incorrect. To avoid these artifacts, the authors need to add "blanks" to their experiments. The authors need to carefully double-check their results. Otherwise, other researchers studying plant metabolites will cite this work to prove that plants can synthesize siloxanes.
Response:
We appreciate your rigorous review and valuable suggestions for our study. Indeed, we failed to fully consider the potential sources of siloxanes in the experimental design previously. In response to your advice, we revisited relevant literature and supplemented control experiments including reagent blanks and instrumental blanks. The results showed that siloxanes were detected in all blank groups, confirming your judgment. Consequently, we have deleted the misleading statements in the original manuscript and will strictly follow methodological norms to design control experiments in future research to avoid similar issues. Thank you again for your input, which has enhanced the rigor of our study.
Your suggestion has greatly enhanced the analysis depth of this article. Thank you again for your professional guidance.
Good luck with your work.
Jianhao Wu
Round 3
Reviewer 2 Report
Comments and Suggestions for Authors
The authors of the manuscript removed one siloxane from the table with the GC-MS analysis results. However, there are still some left: Disiloxane, Silane. These compounds are also present in the discussion of the results.
I would also like to mention the mineral acid: Phosphoric acid. I doubt that it can be determined by the GC-MS method.
The situation with the sulfone remains unclear. Did the authors find some metabolite containing a -SO2 group in its structure? This result needs to be clarified.
Author Response
Dear Reviewer:
Thank you for your comments once again on our manuscript entitled "Metabolomics Analysis Reveals the Influence Mechanism of Different Growth Years on the Growth, Metabolism and Accumulation of Medicinal Components of Bupleurum scorzonerifolium Willd. (Apiaceae)" (biology-3646555). Those comments are all valuable and very helpful for revising and improving our paper, as well as the important guiding significance to our researches. We have carefully studied the comments and corrected them, hoping for approval. The main corrections in the paper and the responds to the reviewer’s comments are as flowing:
Responds to the reviewer’s comments:
Comments:
The authors of the manuscript removed one siloxane from the table with the GC-MS analysis results. However, there are still some left: Disiloxane, Silane. These compounds are also present in the discussion of the results.
Response:
We thank the reviewer for their attention to the siloxane compounds in the GC-MS analysis results. In the previous revision, we had already removed one siloxane compound from the table, and we have now further improved the manuscript as per your suggestion: all remaining entries of disiloxane, silane, etc., have been removed from the GC-MS analysis results table. Meanwhile, the discussion section has been correspondingly revised to delete all discussions related to siloxane compounds, so as to avoid misinterpretation of their biological significance.
Comments:
I would also like to mention the mineral acid: Phosphoric acid. I doubt that it can be determined by the GC-MS method. The situation with the sulfone remains unclear. Did the authors find some metabolite containing a -SO2 group in its structure? This result needs to be clarified.
Response:
In view of the methodological controversies regarding the detection of phosphoric acid and sulfite in plant tissues by GC-MS, we have prudently deleted the descriptions and corresponding data on the presence of phosphoric acid and sulfite in the paper to avoid potential misguidance to related research. It should be noted that these signals were detected in Bupleurum chinense and other multiple test plant samples in our study, and their common occurrence may be related to the sulfur/phosphorus metabolic pathways in plants, environmental factor exposure, or potential interferences in the detection process. In the follow-up, we will carry out in-depth research on this phenomenon, plan to cross-validate the structure and content of the target substances through multiple technical means such as HPLC-MS/MS and ICP-MS, and explore their biological sources and metabolic regulatory mechanisms in combination with environmental factor analysis and omics technologies, so as to clarify the relevant controversies with more rigorous scientific arguments.
Your suggestion has greatly enhanced the analysis depth of this article. Thank you again for your professional guidance.
Good luck with your work.
Jianhao Wu